# Efficacy of Biopesticides in the Management of the Cotton Bollworm, *Helicoverpa armigera* (Noctuidae), under Field Conditions

**DOI:** 10.3390/insects13080673

**Published:** 2022-07-27

**Authors:** Lawrence N. Malinga, Mark D. Laing

**Affiliations:** 1Agricultural Research Council–Industrial Crops, Rustenburg 0300, South Africa; 2South African Sugarcane Research Institute, Mount Edgecombe 4300, South Africa; 3School of Agricultural, Earth and Environmental Sciences, College of Agriculture, Engineering & Science, University of KwaZulu-Natal, Pietermaritzburg 3209, South Africa; laing@ukzn.ac.za

**Keywords:** cotton, *Helicoverpa armigera*, biopesticides, Bolldex^®^, Karate^®^

## Abstract

**Simple Summary:**

Cotton remains the most important cash crop in the world. The key insect pests of cotton include the African bollworm *Helicoverpa armigera*. This pest causes damage to crops estimated at greater than USD 2 billion per year worldwide. Excessive use of insecticides to control this pest has a negative effect on the environment, and is expensive for the farmers. The aim of this study is to explore the field efficacy of different biopesticides as an alternative to control *H. armigera*. Four biopesticides—namely, Eco-Bb^®^ (*Beauveria bassiana*), Bb endophyte (*Beauveria bassiana*), Bolldex^®^ (nucleopolyhedrovirus), and Delfin^®^ (*Bacillus thuringiensis*)—were evaluated and compared with the pyrethroid Karate^®^ (lambda-cyhalothrin) and an untreated control. Plots treated with Karate^®^ had significantly lower numbers of *H. armigera* larvae compared to the untreated control, and were comparable to the plots treated with Bolldex^®^. On average, plots treated with Bolldex^®^ had a high seed cotton yield compared to the other treatments. Biopesticides showed a moderate reduction in the numbers of *H. armigera* larvae, and could thus be used within an integrated pest management programme.

**Abstract:**

Cotton is one of the most valuable materials in the world, popularly used in the clothing industry and other products. However, its production is limited by the high infestation of insect pests. A study was conducted to evaluate the effects of different biopesticides on the control of the African bollworm (*Helicoverpa armigera*) under cotton field conditions. Four biopesticides—namely, Eco-Bb^®^ (*Beauveria bassiana*), Bb endophyte (*Beauveria bassiana*), Bolldex^®^ (Nucleopolyhedrovirus), and Delfin^®^ (*Bacillus thuringiensis*)—were evaluated and compared with the pyrethroid Karate^®^ (lambda-cyhalothrin) and an untreated control against *H. armigera*. Field trials were conducted at the Agriculture Research Council, Rustenburg, in the North West Province of South Africa during the 2017 and 2018 cotton seasons. The results revealed that in plots sprayed with Karate^®^ and Bolldex^®^, the numbers of *H. armigera* were significantly reduced compared to the untreated controls. Plots treated with Bolldex^®^ had the lowest number of damaged bolls in 2017, while those treated with Karate^®^ had the lowest number of damaged bolls in 2018. All treated plots had significantly fewer damaged bolls when compared to the controls. A seed cotton yield of 5987 kg/ha was recorded in the plots that were treated with Bolldex^®^—significantly higher than the yields from plots treated with Eco-Bb^®^, Delfin^®^, and Bb endophyte—in 2017. However, the yield in treatments with Eco-Bb^®^, Delfin^®^, and Bb endophyte was lower than that from the untreated controls during this season. In 2018, plots treated with Bolldex^®^ had the highest yield, at 6818 kg/ha, which was not different from the other treatments. The highest average seed cotton yield of 6400 kg/ha was recorded in the plots treated with Bolldex^®^, followed by Karate^®^. In summary, the efficacy of different biopesticides against *H. armigera* varied significantly, while the synthetic pesticide (Karate^®^) and Bolldex^®^ resulted in more consistent control of this pest. The results suggest that biopesticides may, however, have the potential for use in the sustainable control of cotton bollworms as part of integrated pest management programmes, although further work is required to support this hypothesis.

## 1. Introduction

Cotton, *Gossypium hirsutum* L. (Malvaceae), remains the most important cash crop in the world [1]. It is a fibre crop grown in more than 80 tropical and subtropical countries [2,3]. Over a century ago, cotton contributed more than 70% of the world’s fibre use—compared to about 30% of current consumption [4]. The decline is due to the development of a high number of synthetic fibres, which increased in the mid-1990s [5]. Furthermore, the decline is also attributed to climate change and pest problems, among other factors [6]. Cotton production is a major economic component in Africa, and is a significant contributor to economic growth [7]. The continent contributes approximately 8% of the cotton produced worldwide from a total of six cotton basins [8], covering over 20 countries in sub-Saharan Africa [9]. In South Africa, cotton is a significant crop produced by 250 commercial and over 2000 small-scale farmers in five provinces, namely, KwaZulu-Natal, Limpopo, Mpumalanga, North West, and Northern Cape [10,11].

The key insect pests of cotton include the African bollworm, *Helicoverpa armigera* Hübner (Noctuidae: Lepidoptera). This bollworm is an indigenous species considered to be a major pest of fibre crops in Africa [12], and ranks as the most important lepidopteran pest in South Africa [13,14,15]. Four heliothine species are reported as being of economic importance in Africa, but *H. armigera* is the only species of major economic importance [16]. This pest causes damage to crops estimated at greater than USD 2 billion annually in Asia, Europe, Africa, and Australasia [17]. Unlike most other arthropod pests, *H. armigera* is a polyphagous pest that infests more than 200 crop species worldwide, and can adapt to new environments [18,19]. It has a very large range of host plants, including cotton, pepper, corn, tomato, lucerne, soybean, sorghum, and tobacco [20,21]. *H. armigera* is a serious pest because of its polyphagy [22,23], high fecundity [24], high mobility [25], and resistance to chemical insecticides [26,27,28,29,30]. Cotton is mainly attacked by the larval stage of *H. armigera* [31], which causes high yield loss. Because *H. armigera* larvae have a habit of entering the fruit, bolls, or pods, the plant affords them good protection against chemical sprays, making control almost impossible [32]. Low economic damage thresholds in cotton require a high level of control [12], resulting in reliance on synthetic insecticides [33,34]. 

Worldwide, up to 60% of all commercialised insecticides are used in cotton [35]. Chemical pesticides such as pyrethroids, carbamates, and organophosphates are applied to control *Helicoverpa* pests [36]. Pyrethroids such as Karate^®^ (lambda-cyhalothrin) are non-selective insecticides commonly used against pests—including lepidopterans—in cotton [37,38]. Although chemical pest control is extensively used throughout the world, it has been generally regarded as detrimental to the environment [39]. The use of chemical pesticides to control agricultural pests has resulted in secondary pest outbreaks, as well as reductions in the numbers of beneficial insects and biodiversity [19]. Excessive use of chemicals also causes economic impacts on farmers, and has a negative impact on animals [40]. Before the introduction of Bt cotton with the bacterium *Bacillus thuringiensis* in 1996 [41], cotton production was costly due to the high application of pesticides [42]. However, with the continuous use of Bt cotton on a large scale, *H. armigera* has developed resistance to Bt cotton [43,44]. Today, many agricultural chemical pesticides are under pressure to be banned due to their harmful effects, and some farmers have turned to biopesticides [45,46]. 

Biopesticides are used to control plant pests, and are made from living micro-organisms or natural products [47]. They are known to be economical, environmentally friendly, and target-specific [48]. Biopesticides have the potential to control yield loss without compromising the quality of the crop [49]. The bacterium *Bacillus thuringiensis* is one of the most used microbial biopesticides. Although *Helicoverpa armigera* nucleopolyhedrovirus (HearNPV) is generally used against cotton bollworms, little is understood about the interaction between the virus and host insects [50]. The use of biopesticides in Africa is minimal, accounting for only 3% of the world’s market [51]. South Africa and Kenya are the leading countries in the development and use of biopesticides in Africa [52]. By 2019, South Africa had over 30 registered biopesticides, of which 7 are manufactured locally, and mainly comprise different strains of *Beauveria bassiana* [53].

The idea behind IPM is that combining different practices overcomes the shortcomings of individual practices. The aim is not to eradicate pest populations but, rather, to manage them below levels that cause economic damage [54]. Although biopesticides may not be as rapid as synthetic pesticides, they form part of the crop protection methods used in integrated pest management. Thus, biopesticides are one of the promising alternatives for managing environmental pollution. Integrated control for *H. armigera* that seeks to minimise insecticide use and impacts on non-target organisms needs to be considered. However, in South Africa, there are very few registered biopesticides for use against *H. armigera.* Hence, this study confines itself to evaluating the field efficacy of different biopesticides in the control of *H. armigera*. The importance of this study is to enable scientists to intensify research on the stability of biopesticides under field conditions. This will further provide the bioagents industry with knowledge on biopesticides that have the potential to be registered for use on cotton for the control of *H. armigera.*

## 2. Materials and Methods

### 2.1. Trial Site, Layout, and Planting

The trials were conducted in 2017 and repeated in 2018 at the Agricultural Research Council (25°39.0′ S, 27°14.4′ E) in Rustenburg, North West Province, South Africa. Each plot had 6 rows that were 5 m long, with 1 m spacing between rows, 2 m paths between replications, and 20 cm spacing between plants. The trials were completely randomised block designs, and the treatments were replicated four times. A non-GM cotton cultivar, DeltaOPAL, was hand-planted under irrigated conditions, and weeds were manually removed by hand-hoeing. Thinning of seedlings at the fourth true leaf stage was performed to obtain a plant population density of five plants per metre. The trials were planted late in October. 

### 2.2. Treatments and Application

Based on weekly scouting for bollworms, the administration of treatments started 13 weeks after planting, when the first and second instars were observed. Weekly spray applications were performed until 23 weeks after planting, when the cotton bolls were matured. Ten ground applications were administered around 3–4 p.m. due to the UV sensitivity of the biopesticides [55]. The treatments (Table 1) were applied using knapsack sprayers. 

### 2.3. Data Collection

The efficacy of different treatments was assessed based on in situ counts of *H. armigera* larvae. From 12 weeks after planting, 12 plants per plot were scouted weekly. The scouting was conducted using a visual examination of plants in representative locations within a plot. The whole plant was inspected for the presence of all *H. armigera* larval instars, and the total population was counted. The counts were expressed in insect numbers. The seed cotton yields were determined at the end of the season. For accurate yield measurements, and to minimise experimental error, the two middle rows were harvested in each plot. The trials were harvested in May, when over 90% of the bolls had opened. The two middle rows were harvested per plot. Hand-harvesting was performed to ensure that the seed cotton was harvested and weighed accurately. 

### 2.4. Data Analysis

The data were subjected to appropriate analysis of variance (ANOVA). The Shapiro–Wilk test was performed on the standardised residuals to test for deviations from normality [56]. LSDs (least significant differences) were calculated at a 5% significance level to compare the means of significant source effects [57]. The analysis was performed using Genstat Release 18 and SAS version 9.4 statistical software [58]. The seed cotton yields were expressed in percentages (yield in treatment plots–yield in control plots ÷ yield in control plots × 100).

## 3. Results

In 2017, plots treated with Karate^®^ had significantly lower numbers of *H. armigera* larvae compared to the untreated controls, and were comparable to the plots treated with Bolldex^®^ and Bb endophyte (Figure 1). The controls had the highest numbers of *H. armigera,* and this trend was similar for both the 2017 and 2018 seasons. The *H. armigera* population was significantly lower in all of the treatments compared to the untreated controls during the 2018 season. Plots treated with Karate^®^ had lower numbers of *H. armigera* larvae, which were comparable to the plots treated with Bolldex^®^ and Eco-Bb^®^. The results shown in Figure 2 reveal that plots treated with Bolldex^®^ had significantly lower numbers of damaged bolls compared to Bb endophyte and the controls in 2017. However, in 2018, none of the active treatments were different from one another, although plots treated with Karate^®^ had the lowest numbers of damaged bolls, followed by Eco-Bb^®^ and Bolldex^®^. In both seasons, all of the treatments had significantly lower numbers of damaged bolls when compared to the untreated controls. A seed cotton yield of 5987 kg/ha was recorded in the plots that were treated with Bolldex^®^—significantly higher than the yields from the plots treated with Eco-Bb^®^, Delfin^®^, and Bb endophyte—in 2017 (Table 2). However, none of the treatments showed a significantly higher yield than the untreated controls. In 2018, plots treated with Bolldex^®^ had the highest yield, at 6818 kg/ha, which was not different from the other treatments. The yield from the treatments with Eco-Bb^®^, Delfin^®^, and Bb endophyte was lower than that from the untreated controls in 2017. The average seed cotton yield was higher for all treatments in 2018 than in 2017. On average, the plots treated with Bolldex^®^ had a significantly higher seed cotton yield (45%) compared to the untreated controls, followed by Karate^®^. Plots where Karate^®^ was applied had earlier boll opening than the other treatments in 2017.

## 4. Discussion

In this study, statistically significant control of *H. armigera* and increases in yield were obtained with some of the biopesticides when compared to untreated controls. However, none of the biopesticides performed significantly better than the chemical control in the reduction in the numbers of *H. armigera*. Unlike greenhouse trials under controlled conditions, field trials are affected by different environmental factors [59], which may influence the effectiveness of the biopesticides. This study clearly shows that the insecticidal action of some of the tested biopesticides was comparable to the commercial synthetic insecticide. Bolldex^®^ warrants more attention, because the reduction it caused in the numbers of *H. armigera* larvae was comparable to that of Karate^®^. These results demonstrate the potential of biopesticides to reduce *H. armigera* populations, and that they can be introduced as environmentally friendly pesticides in organic and commercial agriculture. This study demonstrates that both Bolldex^®^ and Karate^®^ significantly reduced *H. armigera* populations during the two seasons. Biopesticides have been reported to effectively reduce the larval population of *H. armigera* when combined with parasitoids [60].

*Helicoverpa armigera* nucleopolyhedrovirus (HaNPV) has been used in several countries and introduced in South Africa for use on several crops [32,61]. Delfin^®^ may have reduced the larval population of *H. armigera* compared to the control in 2018, but there were no significant differences in 2017. Khalique and Ahmed (2001) [62] reported that the mortality response of *H. armigera* larvae to a combination of Karate^®^ and *B. thuringiensis* subsp. kurstaki was synergistic, and that the two products were compatible. While considering the economic injury levels of cotton bollworms per plot [63], the results show that *H. armigera* larvae exceeded the threshold level of five bollworms for all of the treatments except for the Karate^®^ treatment in 2017, and there were less than four bollworms for all of the treatments except for the untreated controls in 2018.

The lowest number of damaged bolls was observed in plots that were treated with Bolldex^®^ and Karate^®^. The effects of Bolldex^®^ and Karate^®^ on the reduction in boll damage corresponded with the reduction in the numbers of *H. armigera* on the plots where these treatments were applied. These results are also in concordance with the observations of Li et al. (2006) [64], who found that more than 60% parasitism of *H. armigera* decreased boll damage by 80% compared with controls. Joubert (2012) [32] reported that a trial was conducted for the control of *H. armigera* on peaches, and Bolldex^®^ yielded 99% scar-free fruit.

The data on yield revealed that a significantly higher yield of seed cotton was recorded in the treatments with Bolldex^®^, followed by Karate^®^, in 2018. In 2017, all of the treatments had yields that were not significantly different from the untreated controls. This may be related to the insect numbers in 2017 being higher than in 2018. In 1997, Cole et al. [65] reported that Karate^®^ increased cotton yield by 12% and provided good pest control whilst maintaining beneficial populations. This is contrary to the findings of Kumar and Stanley (2010) [66], who reported that although lambda-cyhalothrin enhanced seed cotton yields, it caused mortality in both destructive and useful insect species. Sinno et al. (2021) [67] reported that endophytic fungi have the potential to improve plant development and provide protection against pests. However, in this study, the endophytic treatments Eco-Bb^®^ and Bb endophyte only suppressed the *H. armigera* population, but had a lower yield compared to untreated controls in one season. Lotfy and Moustafa (2021) [68] investigated the efficacy of two *B. bassiana* strains against the cotton bollworm, and concluded that *B. bassiana* was efficient against the pest. They further observed that the fungus significantly reduced the numbers of infested bolls. The present study did not evaluate the impact of the pest complex other than the bollworm.

Plots that were treated with Karate^®^ had earlier boll opening than the other treatments in 2017. The additive effects were probably due to multiple mechanisms that affect the pathogens, as opposed to the fewer control mechanisms provided by a single antagonist. Ali (2016) [69] stated that the average number of open bolls/plant is significantly increased by spraying insecticides and salicylic acid.

## 5. Conclusions

*H. armigera* remains a major pest of cotton, and its infestation reduces the yield. The main management tool to control this pest is the application of synthetic pesticides. The overuse of such agents subsequently results in pesticide resistance, and this highlights the importance of IPM strategies that include the use of biopesticides. Although biopesticides are safer to use, there are limitations to their full adoption as a pest management tool. The application of biopesticides under field conditions poses some challenges, as high doses are required for good efficacy. This study provides insight that although Eco-Bb^®^ is currently not registered to control *H. armigera,* it has a suppressive potential against the population of *H. armigera* larvae. However, during the first season, the yield in treatments with Eco-Bb^®^, Delfin^®^, and Bb endophyte was lower than of untreated controls. Bolldex^®^ provided better control of the pest compared to the untreated controls, and it increased the yield. Although the biopesticides were less effective against the *H. armigera* larvae compared to the chemical pesticide, they have the potential to be used in conjunction with synthetic pesticides to delay the development of pesticide resistance. However, further research that focuses on the efficacy and persistence of different doses and other products is required. More research on the production and application of biopesticides is needed in order to integrate these biological products into cotton production. It is also essential for public and private stakeholders to support and promote the research on biopesticides. Equally important is the training of farmers in the use of biopesticides for the rapid adoption of this technology.

## Figures and Tables

**Figure 1 insects-13-00673-f001:**
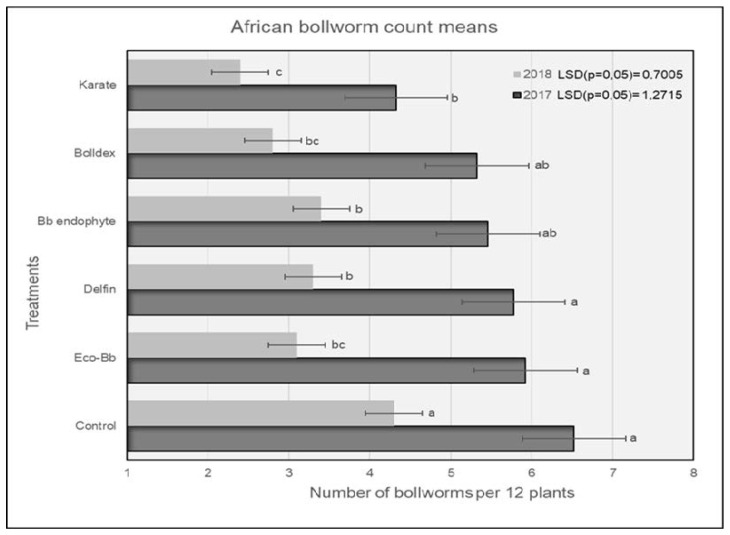
The average number of *H. armigera* larvae found on different treatments during the 2017 and 2018 seasons. Means with the same letter are not significantly different (*p* < 0.05). The comparison was carried out using the data from the same year.

**Figure 2 insects-13-00673-f002:**
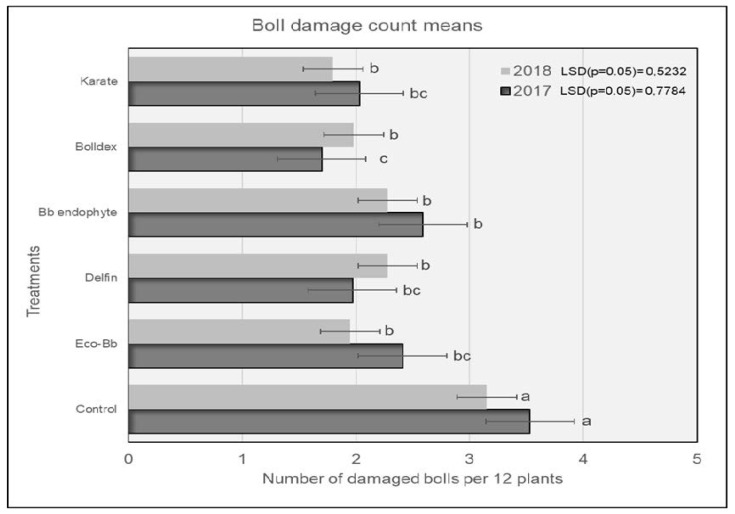
The average number of damaged bolls found on different treatments during the 2017 and 2018 seasons. Means with the same letter are not significantly different (*p* < 0.05). The comparison was carried out using the data from the same year.

**Table 1 insects-13-00673-t001:** List of treatments that were used with their active ingredients, formulations, and concentrations.

Trade Name	Active Ingredient	Formulation	Concentration
Eco-Bb^®^	*Beauveria bassiana* (strain R444)	2 × 10⁹ spores/gram	300 g/ha in 1 g/L water
Bb endophyte	*Beauveria bassiana*	2 × 10⁹ spores/gram	300 g/ha in 1 g/L water
Bolldex^®^	Nucleopolyhedrovirus (HearNPV)	7.5 × 10^12^ spores/gram	200 mL/ha in water
Delfin^®^	*Bacillus thuringiensis* subspecies kurstaki (Btk)	32,000 IU/mg	1 kg/ha in 25 L/ha water
Karate^®^	Lambda-cyhalothrin	50 g/L	120 mL/ha in 200 L/ha water

**Table 2 insects-13-00673-t002:** Seed cotton yields from plots with different treatments during the 2017 and 2018 seasons.

Treatment	2017 (kg/ha) *	2018 (kg/ha) *
Eco-Bb^®^	3055 ± 139.19 b	5961 ± 65.07 ab
Bolldex^®^	5987 ± 86.56 a	6818 ± 95.59 a
Delfin^®^	3523 ± 49.24 b	5755 ± 109. 21 ab
Bb endophyte	3100 ± 66.11 b	6409 ± 128.93 a
Karate^®^	5133 ± 99.34 ab	6405 ± 57.64 a
Untreated control	4168 ± 143.09 ab	4673 ± 124.79 b
LSD (5%)	2373.8	1.6178
CV%	37.94516	17.88032
*p*-Value	0.1216	0.1436

* Means with the same letter are not significantly different (*p* < 0.05). The comparisons were carried out using the data from the same year.

## Data Availability

The data presented in this study are available on request from the corresponding author.

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
