# Peer review of "Efficacy of Biopesticides in the Management of the Cotton Bollworm, Helicoverpa armigera (Noctuidae), under Field Conditions"

_insects, 2022, doi:10.3390/insects13080673_

Round 1

Reviewer 1 Report

An interesting paper on alternative management strategies for H. armigera. More information needs to be included in the introduction and methods. My detailed comments and suggestions are in the attached file. 

Reviewer 2 Report

The manuscript is a decent 1st draft and the science seems sound, but there are too many omissions (mostly in the data collection section) and statements that are not backed up by the data. The introduction is too short and the conclusions section is almost insignificant (4 lines). The authors should go through a significant review of the manuscript before resubmission.

Line 64 – ‘environmentally unacceptable’ replace with ‘detrimental to the environment’

Line 65 – ‘produces harmful side effects on the environment and mammals’ essentially repeats the 1st sentence

Line 67 – ‘and farmers turn to biopesticides’ to ‘and some farmers have turned to biopesticides’

Line 68 – You talk about biopesticides being used as an alternative, but it would be nice to have some examples of the biopesticides being used here, e.g. is it just entomopathogenic fungi? Nematodes?

Line 69 – ‘One of the ways to overcome this situation is to use eco-friendly control measures, like biopesticides’ consider deleting this sentence it essentially repeats the previous sentence and therefore doesn’t make a lot of sense in context.

Overall, the introduction is quite short, I think some background on what biopesticides are and how they are currently being used would add significantly to the paper.

Line 79 – ‘A non-GM cultivar, DeltaOPAL’ A non-GM cultivar of cotton I assume? It’s not actually specified.

Line 82 – ‘in mid-late October.’ Add the year the trial was performed and add precise dates if possible.

Line 86 – ‘done’ replace with ‘performed’.

Line 87 – ‘late afternoon’ please add a specific time frame as ‘late’ is a bit subjective.

Line 90 – I assume an insect count was also performed since it’s in the results?

Line 117-119 – ‘which was much higher than all the other treatments in 2017; and much higher than the untreated control in 2018’ You can’t really say this as ‘much higher’ is very subjective plus the difference was not always statistically significant, for example Bolldex is not significantly different to Karate in either 2017 or 2018.

You also fail to mention in the results that the yield of all treatments in 2017 were not significantly different to the control. This may be related to the insect numbers in 2017 being higher than in 2018, you might consider some kind of statistical analysis comparing just insect numbers with yield of cotton to see what that relationship is.

Line 135 – ‘considerable’ change to ‘statistically significant’, delete ‘consequent’

Line 136 – ‘some of the biopesticides’ change to ‘some of the biopesticides when compared to untreated controls.

Line 136 – ‘However, the chemical control had the best performance on the reduction of the H. armigera.’ Again, not significantly different to the Bolldex treatment so you can’t claim this.

Line 142 – ‘very close’ to ‘comparable’

Line 143 – ‘natural friendly’ to ‘environmentally benign’

Line 147 – Is nuclear polyhedrosis virus the biopesticide or is there another in this combination? The sentence is a bit unclear.

Line 154 – ‘the threshold level of five bollworms’ I’m not sure what this threshold level is? Also is that five boll worms per plant or leaf or soothing else? This detail should really be in the data collection section.

Line 170 – ‘mortalities of’ replace with ‘mortality in’

Line 171 – ‘Plots that were treated with Karate® had earlier boll opening than the other treatments in 2017’ Where is this data? Please add if it’s relevant.

Line 174-175 – ‘average number of open bolls/plants is significantly influenced by spraying insecticides and salicylic acid.’ Influenced how? I’m assuming it increases it?

Line 177 – ‘This study showed that biopesticides caused moderate mortality of H. armigera larvae and thus could be used within an integrated pest management programme.’ This is a bit misleading as some of them did not show a statistical difference in mortality compared to the controls.

Line 178-179 ‘Bolldex® had better control of the pest and increased the yield’ Better control than what? Presumably the biopesticides tested?

Line 179-180 ‘As a possible replacement or in conjunction with synthetic pesticides, the development of resistance could be delayed.’ Pesticide resistance I assume? This point is interesting and deserves far more attention than one line.

The conclusions section is far too brief and is quite misleading at times.

Reviewer 3 Report

In my opinion, the authors should seriously rethink and significantly improve this manuscript. There are many ambiguous or incomprehensible statements in the paper. The conclusion is weak and insufficiently substantiated.

The abstract needs to be rewritten.

Lines 39-42. The main conclusion in the abstract should not be so vague.

Line 56. References are to the articles published in 2007, 2011, 2013. Are there more modern papers on insecticide resistance of this pest?

Line 57. Reference is to the article published in 2007. Is there any up-to-date information on cotton production, as well as a map of the distribution of cotton growing areas in Africa?

This work does not take into account the impact of the pest complex other than the bollworm.

The conclusion needs to be rewritten, since in its current form there is no link to the results of this study. The authors wrote about the fact that Bolldex® had better control of the pest in the methodology of the previous study, which casts doubt on the scientific novelty of this study.

I encourage authors to answer questions, that inevitably arise, in a new version of this manuscript:

1. How were the treatments related to the instar of the larvae? How did you monitor and choose the timing of treatments? There is no information about screening for the flight of the adults, the beginning and mass laying of eggs. At which stage of development was the pest at the time “from 12 weeks after planting”?

2. Why were the treatments started at 13 weeks after planting, while according to your study (https://doi.org/10.1016/j.cropro.2021.105578), treatments were applied to eliminate the populations of the bollworm complex from 9 weeks after planting?

3. Why was only one chemical insecticide used in this study, and why was Karate® chosen?

4. What were the prerequisites for the selection of biological insecticides used in this study?

5. What is known about Helicoverpa armigera resistance to chemical and biological pesticides?

6. Is Bt-cotton used in the experiment area? Why was non-GM cotton cultivar DeltaOPAL chosen?

Round 2

Reviewer 2 Report

The paper looks much better, I only have a few minor revisions.

Line 102 – ‘manufacture local’ to ‘manufactured locally’

Line 103 – ‘mainly based on Beauveria bassiana’ to ‘mainly different strains of Beauveria bassiana’

Line 104 –‘IPM’ to ‘Integrated Pest Management (IPM)‘ and the definition is a bit weak consider revising.

Line 159 – ‘low’ to ‘lower’

Line 162 – ‘According to Figure 2, in 2018, none of the treatments’ to ‘However in 2018, none of the active treatments’

Line 163 – ‘however’ to ‘although’

Line 205 - delete ‘5 different’

Line 239 – ‘integrated pest management’ to ‘IPM’

Reviewer 3 Report

The scientific novelty of the research is very weakly indicated in the manuscript. The seed cotton yield in treatments with Eco-Bb®, Delfin®, and Bb endophyte is lower than in untreated control in 2017. In 2018, trends are not reproduced. For reliable statistics, you need more repetitions. The level of the manuscript does not meet the high standards of Insects. In addition, the paper is not ready for publication in any other scientific journal.

It is necessary to check the references, since 66 sources are cited in the text, and 67 ones are in the list.

Some sentences are poorly connected to each other, for example, the last sentence is not a logical continuation of the meaning of the previous one on lines 109-112.

The conclusions do not indicate which biopesticides (Line 237-238 some biopesticides), according to the results of the study, are recommended for the control of Helicoverpa armigera, except Bolldex® for which high effectiveness is established since long.

Round 3

Reviewer 3 Report

Authors demonstrate that "The yield in treatments with Eco-Bb®, Delfin®, and Bb endophyte was lower than in untreated control in 2017". This example suggests that absence of treatment significantly reduces costs without sacrificing yield. However, authors try to convince the reader that biological treatments are more beneficial than chemical treatments, despite the cost and negative effect on yield compared to untreated controls. I can agree with the authors that there may not be enough time for any tasks within the framework of any study, and the report should describe in detail what has been done and what remains to be done within the framework of a possibly other project. However, for a scientific article it is necessary to use only verified and confirmed research results. Perhaps, for this article, the authors will be able to carry out the necessary research later. At present, the conclusion is not supported by reliable results. As I wrote in a previous review: “For reliable statistics, you need more repetitions”. I believe that the article has a major fundamental flaw that cannot be corrected without additional research.
